# Scalar-potential mapping of the steady-state magnetosheath model

Yasuhito Narita[1], Simon Toepfer[1], and Daniel Schmid[2]

[1]Institut für Theoretische Physik, Technische Universität Braunschweig, Mendelssohnstr. 3, 38106 Braunschweig, Germany
[2]Space Research Institute, Austrian Academy of Sciences, Schmiedlstr. 6, 8042 Graz, Austria

**Correspondence:** Y. Narita (y.narita@tu-braunschweig.de)

**Abstract.** The steady-state magnetosheath model has various applications for studying the plasma and magnetic field profile around the planetary magnetospheres. In particular, the magnetosheath model is analytically obtained by solving the Laplace equation for parabolic boundaries (bow shock and magnetopause). We address the question "How can we utilize the magnetosheath model by transforming into a more general, empirical, non-parabolic magnetosheath geometry?" To achieve the goal, we develop the scalar-potential mapping method which provides a semi-analytic estimate of steady-state flow velocity and magnetic field in the empirical magnetosheath domain. The method makes use of a coordinate transformation from the empirical magnetosheath domain into the parabolic magnetosheath domain, and evaluate a set of variables (shell variable and connector variable) to utilize the solutions of Laplace equation obtained for the parabolic magnetosheath domain. Our model uses two invariants of transformation: the zenith angle within the magnetosheath with respect to the direction to the Sun and the ratio of the distance to the magnetopause to the thickness of magnetosheath along the magnetopause-normal direction. The use of magnetopause-normal direction makes a marked difference from the earlier model construction using the radial direction as reference. The plasma flow and magnetic field can be determined as a function of the upstream condition (flow velocity or magnetic field) in a wide range of zenith angle. The scalar-potential mapping method is computationally inexpensive, using analytic expressions as much as possible, and is applicable to various planetary magnetosheath domains.

## 1 Introduction

Steady-state plasma flow and magnetic field can be regarded as a realization of potential field in the planetary magnetosheath region when the vorticity and the electric current are ignored. In such a case, the potential is obtained by solving the Laplace equation, which was elegantly and analytically solved by Kobel and Flückiger (1994) for a parabolic shape of magnetosheath (hereafter KF). The KF potential was further extended to the stream function in the magnetosheath by Guicking et al. (2012). The KF solution made a series of breakthroughs in the magnetosheath research. One of the most successful applications is the ability to track the plasma parcel along the streamline in the modeled magnetosheath. The tracking method was extensively used to observationally study the mirror mode growth (e.g., Tatrallyay et al., 2002; Génot et al., 2011) and the streamwise turbulence evolution in the magnetosheath (Guicking et al., 2012). Predictive models of plasma flow and magnetic field serve as a useful tool when combined with the numerical simulation or the observational data.

The KF potential is obtained using the assumption that the planetary bow shock and magnetopause have a parabolic shape sharing the same focal point. Empirical models of the bow shock and magnetopause (fitted to the spacecraft data), on the other

hand, are not necessarily parabolically or co-focally shaped. For example, the empirical Earth bow shock model by Farris et al. (1991) and Cairns et al. (1995) has a parabolic shape but the focal point differs from that of the KF solution. The empirical magnetopause model by Shue et al. (1997) applies a power-law scaling to the parabolic shape such that the magnetic field lines appear stretched in the tail region. The gap between the KF parabolic magnetosheath and the empirical magnetosheath needs to be filled when applying the KF potential in the empirical magnetosheath.

Naively speaking, one wishes to find a conformal mapping (angle-preserving mapping) from the KF parabolic magnetosheath onto a non-parabolic empirical magnetosheath shape such as the analytic extension of magnetopause shape (Narita et al., 2023). However, no general mathematical algorithm is known so far to obtain the conformal mapping when the spatial domain is not properly bounded. The problem lies in the fact that the magnetosheath is bounded only by two sides, i.e., the standing shock and the magnetopause in the radial direction to the planet, but not bounded along the flow in the tail region. The algorithms of numerical conformal mapping are so far proposed for spatially bounded domains (Papamichael and Whiteman, 1973; Chakravarthy and Anderson, 1979; Fornberg, 1980; Karageorghis et al., 1996) or domains with a closed shape of internal boundaries (Wei et al., 2014).

Here we addresss the question "How can we utilize the KF magnetosheath model by transforming into a more general, empirical, non-parabolic magnetosheath geometry?" To achieve the goal, we develop a mapping method which provides a semi-analytic estimate of steady-state flow velocity and magnetic field in the empirical magnetosheath domain. Our scalar-potential mapping method is computationally inexpensive by using the analytic expression as much as possible, and is applicable to various planetary magnetosheath domains.

This work is organized in the following fashion. After reviewing the magnetosheath model constructed by Kobel and Flückiger (1994) (section 2 and discussing different mapping methods (section 3), the detailed procedure of the magnetopause-normal mapping is presented (section 4) with concluding remark (section 5).

## 2 Revisiting the magnetosheath scalar potential

### 2.1 Parabolic coordinates

In the KF parabolic coordinates, the shell variable $v$ (iso-contour lines enveloping the magnetosphere) and the connector $u$ (iso-contour lines connecting from the bow shock to the magnetopause) play an important role in computing the flow velocity and magnetic field in the magnetosheath. These variables are explicitly evaluated using Cartesian coordinates and the radial distance from the focal point as

$$v = \sqrt{r_0 + (x_\mathrm{k} - x_0)} \tag{1}$$
$$u = \sqrt{r_0 - (x_\mathrm{k} - x_0)}, \tag{2}$$

where $r_0$ is the distance to the focus at $x_0$:

$$r_0 = \sqrt{(x_\mathrm{k} - x_0)^2 + y_\mathrm{k}^2 + z_\mathrm{k}^2}. \tag{3}$$

The focus is along the x axis, and is defined as

$$x_0 = \frac{1}{2} R_{\mathrm{mp}}. \tag{4}$$

$x_{\mathrm{k}}$, $y_{\mathrm{k}}$, and $z_{\mathrm{k}}$ are the Cartesian representation of the KF magnetosheath model (i.e., with the pre-fixed bow shock and magnetopause shapes) obtained by projecting the position vector onto the unit vectors $\boldsymbol{e}_x$, $\boldsymbol{e}_y$, and $\boldsymbol{e}_z$:

$$x_{\mathrm{k}} = \boldsymbol{r}^{(\mathrm{k})} \cdot \boldsymbol{e}_x \tag{5}$$

$$y_{\mathrm{k}} = \boldsymbol{r}^{(\mathrm{k})} \cdot \boldsymbol{e}_y \tag{6}$$

$$z_{\mathrm{k}} = \boldsymbol{r}^{(\mathrm{k})} \cdot \boldsymbol{e}_z. \tag{7}$$

To complete the variable set for computing the potentials and the stream function, the azimuthal angle $\phi$ is introduced as

$$\phi = \mathrm{atan}(z_{\mathrm{k}}/y_{\mathrm{k}}). \tag{8}$$

## 2.2 Velocity potential

In the frame of potential field theory, the flow velocity $\boldsymbol{U}$ is obtained either from the velocity potential (a scalar potential) $\Phi^{(\mathrm{vel})}$ or from the stream function (also a scalar potential) $\Psi$ as

$$\boldsymbol{U} = -\nabla \Phi^{(\mathrm{vel})} = -\nabla \times \left( \Psi \boldsymbol{e}_\phi \right). \tag{9}$$

The symbol $\boldsymbol{e}_\phi$ is the unit vector in the azimuthal direction around the symmetry axis (Sun-to-planet direction). Kobel and Flückiger (1994) and Guicking et al. (2012) obtained the analytic expression of the velocity potential $\Phi^{(\mathrm{vel})}$ using the shell variable $v$ and the connector variable $u$.

$$\begin{aligned} \Phi^{(\mathrm{vel})} &= -U_x \left( \frac{v_{\mathrm{mp}}^2 v_{\mathrm{bs}}^2}{v_{\mathrm{bs}}^2 - v_{\mathrm{mp}}^2} \right) \left( \frac{u^2 - v^2}{2 v_{\mathrm{bs}}^2} + \ln v \right) - \\ & \quad \frac{1}{2} U_x \left( u^2 - v^2 \right) + \Phi_0^{(\mathrm{vel})}, \end{aligned} \tag{10}$$

where $U_x$ is the upstream flow velocity, $v_{\mathrm{mp}}$ the shell variable at the magnetopause, $v_{\mathrm{bs}}$ the shell variable at the bow shock, $v$ the shell variable, $u$ the connector variable, and $\Phi_0^{(\mathrm{vel})}$ a free parameter (integration constant) which is set to zero without loss of generality. The boundary shell values $v_{\mathrm{mp}}$ and $v_{\mathrm{bs}}$ contain the information on the stand-off distances ($R_{\mathrm{mp}}$ and $R_{\mathrm{bs}}$) in the subsolar region, and are defined by Kobel and Flückiger (1994) as

$$v_{\mathrm{mp}} = \sqrt{R_{\mathrm{mp}}} \tag{11}$$

$$v_{\mathrm{bs}} = \sqrt{2 R_{\mathrm{bs}} - R_{\mathrm{mp}}}. \tag{12}$$

## 2.3 Stream function

Guicking et al. (2012) transformed the KF potential and obtained analytically the stream function $\Psi$ as a function of the shell variable and the connector variable:

$$\Psi = -\frac{1}{2}U_x\left(\frac{v_{\mathrm{mp}}^2 v_{\mathrm{bs}}^2}{v_{\mathrm{bs}}^2 - v_{\mathrm{mp}}^2}\right)\frac{u}{v}\left(\frac{v^2}{v_{\mathrm{bs}}^2} - 1\right) - \frac{1}{2}U_x uv. \tag{13}$$

Hereafter, one may set $U_x = -1$ so that the velocity potential $\Phi^{(\mathrm{vel})}$ is normalized to the upstream velocity. Isocontour lines of the stream function represent the streamline.

## 2.4 Magnetic scalar potential

The magnetic field in the magnetosheath is derived from the scalar potential in the same fashion as the flow velocity, that is,

$$\boldsymbol{B} = -\nabla\Phi^{(\mathrm{mag})}. \tag{14}$$

The magnetic potential is a function of the shell variable $v$ and the connector $u$ (Kobel and Flückiger, 1994):

$$\begin{aligned}
\Phi^{(\mathrm{mag})} = & -\frac{v_{\mathrm{mp}}^2 v_{\mathrm{bs}}^2}{v_{\mathrm{bs}}^2 - v_{\mathrm{mp}}^2} \times \\
& \left[\left(B_y^{(\mathrm{up})}\cos\phi + B_z^{(\mathrm{up})}\sin\phi\right)u\left(\frac{1}{v} + \frac{v}{v_{\mathrm{bs}}^2}\right) + \right. \\
& \left. B_x^{(\mathrm{up})}\left(\frac{u^2 - v^2}{2v_{\mathrm{bs}}^2} + \ln v\right)\right] \\
& - B_x^{(\mathrm{up})}(-x) - B_y^{(\mathrm{up})}y - B_z^{(\mathrm{up})}z + \Phi_0^{(\mathrm{mag})},
\end{aligned} \tag{15}$$

where $B_x^{(\mathrm{up})}$ is the sunward component of the upstream magnetic field (corresponding to the GSE-X in near-Earth space), and $B_y^{(\mathrm{up})}$ and $B_z^{(\mathrm{up})}$ are two components of the upstream magnetic field perpendicular to the x direction. $\phi$ is the azimuthal angle of the position around the symmetry axis (the y direction is given by the angle $\phi = 0$). The integration constant is chosen as $\Phi_0^{(\mathrm{mag})} = 0$. The magnetic potential cannot be further transformed into the form of stream function since the magnetic field distribution is essentially three-dimensional in the magnetosheath.

## 3 Mapping method comparison

### 3.1 Mapping problem

Our task is to find the shell variable $v$ and the connector $u$ in the empirical magnetosheath by finding a suitable mapping of the position vector from the empirical magnetosheath (denoted by $\boldsymbol{r}$) onto the KF parabolic system (denoted by $\boldsymbol{r}^{(\mathrm{k})}$). The evaluated $v$ and $u$ are then readily used to obtain the scalar potentials and the stream function. The flow velocity and the magnetic field in the empirical magnetosheath are obtained by computing the gradient of the respective potential.

A practically useful mapping procedure to utilize the KF potential is proposed by Soucek and Escoubet (2012) by using the radial direction as a reference. While the radial mapping can reasonably (i.e., with a relatively high accuracy) transform the KF potential into the empirical magnetosheath domain on the dayside, the mapping quality becomes degraded in the flank region due to the conversion effect associated with the non-orthogonal grid construction. Our approach differs from the radial mapping by using the magnetopause-normal direction as a reference. The azimuthal coordinate $\phi$ is still orthogonal to the $u$ and $v$ coordinates. We briefly compare between the two mapping methods here.

### 3.2 Radial direction as reference

Soucek and Escoubet (2012) presented in their pioneering work an algorithm of radial mapping by transforming the KF magnetosheath model into a general, empirical magnetosheath shape by referring to the radial direction from the planet and scaling the radial position in the magnetosheath to the KF model. While the radial mapping can reasonably (i.e., with a relatively high accuracy) transform the KF potential into the empirical magnetosheath domain on the dayside, the mapping quality becomes degraded in the flank region due to the strongly non-orthogonal grids. Figure 1 displays a comparison of the radial grids between the KF magnetosheath model and the empirical magnetosheath model. The grids span the radial direction to the planet and transfinite interpolation between the bow shock and the magnetopause. The radial mapping has a drawback in a stronger grid non-orthogonality effect, which causes an artificial converging flow pattern in the flank region (velocity potential shown in Fig. 2 when the scalar potential is directly transformed. In the Soucek-Escoubet method, the problem of flow conversion effect was avoided by solving the MHD Rankine-Hugoniot relation and tracking the streamline iteratively between the KF parabolic magnetosheath model and the empirical magnetosheath model.

Although the method introduced by Génot et al. (2011) and later adapted by Soucek and Escoubet (2012) is computational less expensive than global magnetosheath simulations, the density and velocity fields from the bow shock to a given point in the magnetosheath still needs to be computed along the streamline in an incremental way. Moreover, the Rankine-Hugoniot relations need to be solved before calculating iteratively the streamline, the flow velocity vector, to track the plasma density flow velocity along the streamline. Naturally, the uncertainty in this calculations depends on the step size (larger uncertainties for larger step sizes) and the errors accumulate along the streamline. The method introduced in this work overcomes this issue by constructing a magnetopause-normal grid system such that computational efforts are improved (no need to solve the Rankine-Hugoniot relations and the error does not accumulate in the flank region).

### 3.3 Magnetopause-normal direction as reference

Our mapping method differs from the radial mapping method in that the magnetopause-normal direction is used as a reference to the magnetopause. Our method guarantees the grid orthogonality around the magnetopause both on the dayside and in the flank region. The magnetopause-normal grids are shown in Fig. 3 for the KF magnetosheath model (with parabolic boundaries) and the empirical magnetosheath model (with non-parabolic boundaries). Even though the exact conformal mapping is not available, the magnetopause-normal mapping method retains the grid orthogonality around the magnetopause. This feature

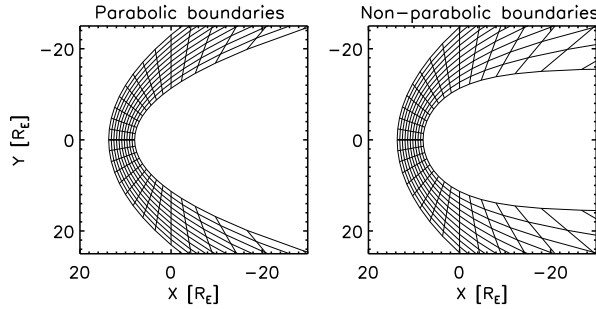

**Figure 1.** Grid pattern generated by the radial mapping for the Kobel-Flückiger parabolic magnetosheath (left panel) and the non-parabolic, empirical magnetosheath (right panel).

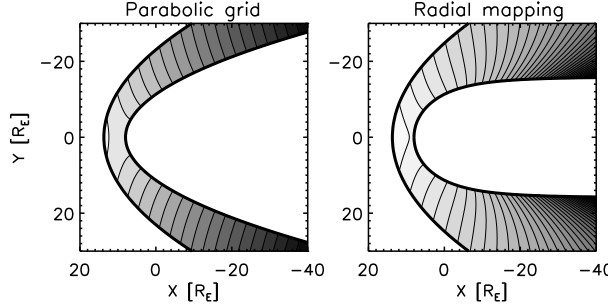

**Figure 2.** Velocity potential in the Kobel-Flückiger model (left panel) and its radial mapping onto the non-parabolic empirical boundaries (right panel). Note that the same function is plotted here for different mapping methods. Contours represent the velocity potential normalized to the solar wind, $\Phi^{(\text{vel})}/U_x$, which is in units of the planetary radii. The color range is chosen for the visual demonstration purpose from $0.2R_{\text{E}}$ to $90R_{\text{E}}$ (left panel) and from $2R_{\text{E}}$ to $200R_{\text{E}}$ (right panel).

(orthogonality around the magnetopause) plays a crucial role in mapping the scalar potentials. An example of the scalar-potential mapping by referring to the magnetopause-normal direction (our final results) are shown in section 4.8.

## 4 Magnetopause-normal mapping

### 4.1 Overview of the procedure

The magnetopause-normal mapping is performed with two transformations. In the first transformation, the position vector is mapped from the empirical magnetosheath $r$ onto the KF magnetosheath model $r_{\text{k}}$. This is based on the assumption that the distance to the magnetopause along the magnetopause-normal direction when scaled to the magnetosheath thickness (defined as the distance from the magnetopause to the bow shock along the magnetopause-normal direction) remains constant. The azimuthal angle $\phi$ is the same between the empirical magnetosheath and the KF model. The first transformation is divided into

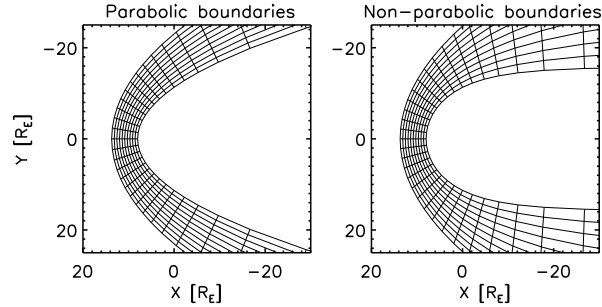

**Figure 3.** Mesh pattern used in the magnetopause-normal mapping in this work for the parabolic boundaries (left panel) and the non-parabolic, empirical boundaries (right panel).

computing the distance to the magnetopause (step 1), the thickness of the empirical magnetosheath (step 2), the thickness of the KF magnetosheath (step 3), and the mapping of the position vector onto the KF model(step 4).

In the second transformation, the mapped position vector is used to compute the shell variable $v$ and the connector variable $u$ (step 5) and to obtain the potentials and the stream function in the empirical magnetosheath using Eqs. (10), (13), and (15) (step 6). Here again, the azimuthal angle $\phi$ is treated as the same.

Figure 4 illustrates the mapping procedure and graphically explains the variables that need to be determined to perform the mapping such as the zenith angle $\theta_{\mathrm{mp}}$ associated with the mimimum distance to the magnetopause the distance from the planet to the bow shock $r_{\mathrm{bs}}$, the distance from the planet to the magnetopause $r_{\mathrm{mp}}$, the relative distance to the magnetopause $\alpha_{\mathrm{emp}}$, and the magnetosheath thickness $\alpha_{\mathrm{emp}}^{(\mathrm{bs})}$. The position vector $\boldsymbol{r}$, the bow shock stand-off distance $R_{\mathrm{bs}}$, the bow shock shape, the magnetopause stand-off distance $R_{\mathrm{mp}}$, and the magnetopause shape are assumed to be known in our mapping.

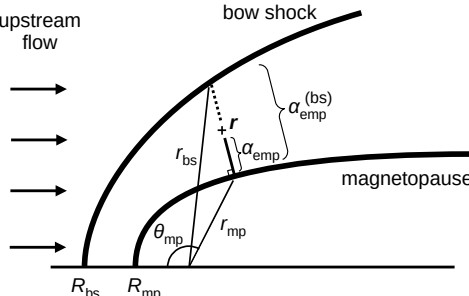

**Figure 4.** Variables used in the magnetopause-normal mapping with the zenith angle $\theta_{\mathrm{mp}}$ along the direction nearest to the magnetopause, the radial distance to the bow shock and magnetopause along the magnetosheath-normal direction ($r_{\mathrm{bs}}$ and $r_{\mathrm{mp}}$, respectively), the distance from the magnetosheath to the magnetopause $\alpha_{\mathrm{emp}}$, the magnetopause thickness $\alpha_{\mathrm{emp}}^{(\mathrm{bs})}$. The position vector is denoted by $\boldsymbol{r}$. The bow shock and magnetopause stand-off distances are denoted by $R_{\mathrm{bs}}$ and $R_{\mathrm{mp}}$), respectively.

## 4.2 Setup

We begin with a position vector in the empirical magnetosheath domain, and express the position vector as $\boldsymbol{r} = x\boldsymbol{e}_x + y\boldsymbol{e}_y + z\boldsymbol{e}_z$. Hereafter, we present the mapping procedure in the two-dimensional plane spanning the x and y directions for simplicity, but the computation in three dimensions is straightforward by representing the y component of position vector in the cylindrical fashion as $\rho\cos\phi$ and the z component into $\rho\sin\phi$ using the distance $\rho$ to the x axis. The boundaries (bow shock and magnetopause) are specified by the users and do not need to be parabolic. In this paper, we use the following bow shock and magnetopause

models.

- The empirical bow shock position expressed in GSE (Geocentric Solar Ecliptic) coordinates proposed and discussed by Farris et al. (1991); Cairns et al. (1995)

$$x = R_{\mathrm{bs}} - b_{\mathrm{emp}}\, y^2, \tag{16}$$

where $R_{\mathrm{bs}}$ is the bow shock stand-off distance and $b_{\mathrm{emp}}$ is the empirical flaring parameter. We note here that the original

Farris empirical bow shock model is not a paraboloid model, it is an ellipsoid model (with an eccentricity of 0.81), describing the bow shock on the dayside. It is not a proper representation of the far flank bow shock. Also, the Cairns paraboloid bow shock model does not properly represent the far flank bow shock. The distant bow shock shape approaches that of a hyperboloid.

- The empirical magnetopause position by Shue et al. (1997):

$$x^2 + y^2 - \frac{4R_{\mathrm{mp}}^4}{4R_{\mathrm{mp}}^2 - y^2} = 0, \tag{17}$$

in the Cartesian representation and

$$r_{\mathrm{mp}} = R_{\mathrm{mp}}\sqrt{\frac{2}{1+\cos\theta}}. \tag{18}$$

in the polar representation.

We use a specific exponent for the Shue model (with an alpha exponent of 0.5) in an effort to show that the analytic model

is 'simple'. The solar wind conditions for which this exponent is applicable is not often encountered (e.g., interplanetary magnetic field has the Bz component larger than +8 nT, with specific values of solar wind dynamic pressure).

In our setup, the radial distance from the planet to the bow shock is expressed as (see appendix)

$$r_{\mathrm{bs}} = \frac{1}{2b_{\mathrm{emp}}\sin^2\theta}\left(-\cos\theta + \right.$$
$$\left. \sqrt{1 - (1 - 4b_{\mathrm{emp}}R_{\mathrm{bs}})\sin^2\theta}\right). \tag{19}$$

The radial distance to the magnetopause is given conveniently by Eq. (18). The Shue model reproduces the magnetopause stand-off distance $R_{\mathrm{mp}}$ in the subsolar direction ($\theta = 0$), and the cylindrical distance asymptotes to $2R_{\mathrm{mp}}$ in the tail. It is worth noting here that one needs to compute the radial distance from the planet to the bow shock or magnetopause as a function of the zenith angle when using different shapes.

### 4.3 Step 1: Measuring the distance to magnetopause

In the first step, the distance from the given position in the magnetosheath to the nearest magnetopause is computed (see Fig. 5). We express the position vector along the magnetopause-normal direction as

$$\boldsymbol{r} = \boldsymbol{r}_{\mathrm{mp}} + \alpha_{\mathrm{emp}} \, \boldsymbol{e}_{\mathrm{mp}}, \tag{20}$$

where $\boldsymbol{r}_{\mathrm{mp}}$ is the magnetopause position nearest to the position vector, and $\boldsymbol{e}_{\mathrm{mp}}$ is the unit vector in the magnetopause-normal direction. The unit vector points away from the planet and satisfies the condition

$$\boldsymbol{r}_{\mathrm{mp}} \cdot \boldsymbol{e}_{\mathrm{mp}} > 0 \tag{21}$$

The symbol $\alpha_{\mathrm{emp}}$ is the distance to the magnetopause along the magnetopause-normal direction $\boldsymbol{e}_{\mathrm{mp}}$ in the empirical magnetosheath.

The nearest magnetopause position is obtained by searching for the zenith angle $\theta_{\mathrm{mp}}$ for the minimum distance from the sample position to the magnetopause. The distance $D$ is defined as

$$D = \sqrt{(r_x - r_{\mathrm{mp}} \cos\theta_{\mathrm{mp}})^2 + (r_y - r_{\mathrm{mp}} \sin\theta_{\mathrm{mp}})^2}. \tag{22}$$

The search for the minimum distance is implemented in a brute-force fashion as a function of $\mu_{\mathrm{mp}} = \cos\theta_{\mathrm{mp}}$ in our study.

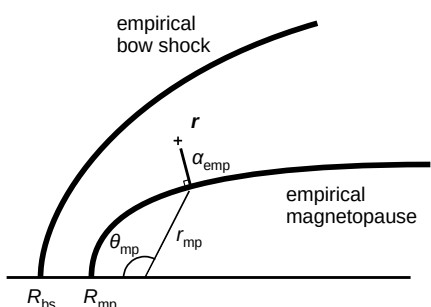

**Figure 5.** Measuring the distance to the empirical magnetopause (Step 1).

Using the minimum distance to the magnetopause $r_{\mathrm{mp}}$ and the zenith angle $\theta_{\mathrm{mp}}$, we are ready to compute the magnetopause-normal direction and the distance $\alpha_{\mathrm{emp}}$. To obtain the magnetopause-normal direction, we define the magnetopause shape function $f_{\mathrm{mp}}$ as

$$f_{\mathrm{mp}} = x^2 + y^2 - \frac{4R_{\mathrm{mp}}^4}{4R_{\mathrm{mp}}^2 - y^2}, \tag{23}$$

and compute the normal direction by the gradient of $f_{\text{mp}}$ as

$$\frac{\partial f_{\text{mp}}}{\partial x} = 2x \tag{24}$$

$$\frac{\partial f_{\text{mp}}}{\partial y} = 2y \left[ 1 - \frac{4R_{\text{mp}}^4}{(4R_{\text{mp}}^2 - y^2)^2} \right]. \tag{25}$$

The magnetopause-normal direction is obtained by normalizing the gradient vector $(\partial_x f_{\text{mp}}, \partial_y f_{\text{mp}})$ and representing with the basis vectors ($\boldsymbol{e}_x$ and $\boldsymbol{e}_y$) as

$$\boldsymbol{e}_{\text{mp}} = \frac{\text{sgn}}{\sqrt{(\partial_x f_{\text{mp}})^2 + (\partial_y f_{\text{mp}})^2}} \times$$
$$(\partial_x f_{\text{mp}} \, \boldsymbol{e}_x + \partial_y f_{\text{mp}} \, \boldsymbol{e}_y) \tag{26}$$

evaluated at the magnetopause ($x = r_{\text{mp}} \cos\theta_{\text{mp}}$ and $y = r_{\text{mp}} \sin\theta_{\text{mp}}$). The magnetopause-normal vector $\boldsymbol{e}_{\text{mp}}$ has a unit length, and the sign ($\text{sgn} = \pm 1$) is chosen such that the normal vector is pointing outward (Eq. 21). The distance $\alpha_{\text{emp}}$ to the magnetopause along the normal direction is obtained from Eq. (20) as

$$\alpha_{\text{emp}} = \frac{(x - r_{\text{mp}} \cos\theta_{\text{mp}}) + (y - r_{\text{mp}} \sin\theta_{\text{mp}})}{\boldsymbol{e}_{\text{mp}} \cdot \boldsymbol{e}_x + \boldsymbol{e}_{\text{mp}} \cdot \boldsymbol{e}_y} \tag{27}$$

Equation (27) is constructed to be robust against the singular behavior on the dayside ($\boldsymbol{e}_{\text{mp}} \cdot \boldsymbol{e}_y = 0$) and in distant tail ($\boldsymbol{e}_{\text{mp}} \cdot \boldsymbol{e}_x = 0$).

## 4.4 Step 2: Computing the thickness of empirical magnetosheath

In the second step, the magnetosheath thickness is computed using the position vector and the magnetopause normal direction (Fig. 6). For our mapping purpose, the distance $\alpha_{\text{emp}}$ is normalized to the magnetosheath thickness $\alpha_{\text{emp}}^{(\text{bs})}$ such that the relative distance $\alpha_{\text{emp}}/\alpha_{\text{emp}}^{(\text{bs})}$ serves as an invariant of the mapping from the empirical magnetosheath onto the KF magnetosheath. To achieve this, we combine Eq. (16) with Eq. (20), and analytically determine the thickness from the bow shock to the magnetopause in the empirical magnetosheath. That is, the thickness $\alpha_{\text{emp}}^{(\text{bs})}$ is obtained by rewriting the bow shock quadratic equation (Eq. 16) for the position vector using the variable $\alpha_{\text{emp}}^{(\text{bs})}$ (Eq. 20) extended to the bow shock location. The equation is again quadratic, and the solution is algebraically obtained as:

$$\alpha_{\text{emp}}^{(\text{bs})} = \frac{1}{2b_{\text{emp}} e_{\text{mp},y}^2} \times$$
$$\left[ -(e_{\text{mp},x} + 2b_{\text{emp}} y_{\text{mp}} e_{\text{mp},y})^2 + d_\alpha \right], \tag{28}$$

where $d_\alpha$ is an auxiliary variable defined as

$$d_\alpha = [(e_{\text{mp},x} + 2b_{\text{emp}} y_{\text{mp}} e_{\text{mp},y})^2 -$$
$$4b_{\text{emp}} e_{\text{mp},y}^2 \times$$
$$(x_{\text{mp}} + b_{\text{emp}} y_{\text{mp}}^2 - R_{\text{bs}})]^{1/2}. \tag{29}$$

In the subsolar direction ($y_{\mathrm{mp}} = 0$), the thickness is simply given as

$$\alpha_{\mathrm{emp}}^{(\mathrm{bs})} = R_{\mathrm{bs}} - R_{\mathrm{mp}}. \tag{30}$$

Equation (28) becomes singular in the subsolar direction and Eq. (30) needs to be set separately.

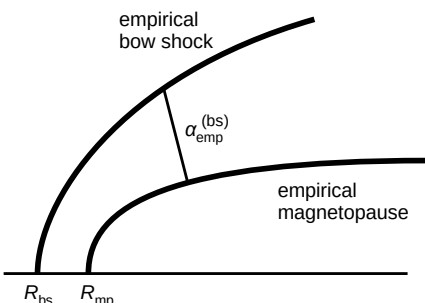

**Figure 6.** Computing the magnetosheath thickness in the empirical model (Step 2).

### 4.5   Step 3: Computing the magnetosheath thickness in the KF system

In the third step, the magnetosheath thickness is computed in the KF model (Fig. 7). We repeat the procedures of steps 1 and 2 for the KF system and determine the KF magnetosheath thickness as reference. We treat the zenith angle $\theta_{\mathrm{mp}}$ and the relative distance $\alpha_{\mathrm{emp}}/\alpha_{\mathrm{emp}}^{(\mathrm{bs})}$ as invariants of the mapping between the empirical magnetosheath and the KF system. The KF bow shock
location is given as

$$x = R_{\mathrm{bs}} - b_{\mathrm{k}} y^2, \tag{31}$$

where the KF bow-shock flaring parameter $b_{\mathrm{k}}$ is pre-fixed as (Kobel and Flückiger, 1994)

$$b_{\mathrm{k}} = \frac{1}{4R_{\mathrm{bs}} - 2R_{\mathrm{mp}}}. \tag{32}$$

The radial distance from the planet to the KF bow shock is

$$r_{\mathrm{bs}}^{(\mathrm{k})} = \frac{1}{2b_{\mathrm{k}} \sin^2 \theta} \times$$
$$\left( -\cos\theta + \sqrt{1 + (4b_{\mathrm{k}} R_{\mathrm{bs}} - 1) \sin^2 \theta} \right). \tag{33}$$

The KF magnetopause is defined in Kobel and Flückiger (1994) as

$$x = R_{\mathrm{mp}} - \frac{1}{2R_{\mathrm{mp}}} y^2. \tag{34}$$

From Eq. (34) the radial distance from the planet to the KF magnetopause is computed as

$$r_{\mathrm{mp}}^{(\mathrm{k})} = \frac{R_{\mathrm{mp}}}{\sin^2 \theta} \left( -\cos\theta + \sqrt{1 + \sin^2 \theta} \right). \tag{35}$$

To obtain the magnetopause-normal direction in the KF system, we compute the gradient of the magnetopause shape function:

$$f_{\mathrm{mp}}^{(\mathrm{k})} = x - R_{\mathrm{mp}} + \frac{1}{2R_{\mathrm{mp}}} y^2. \tag{36}$$

The gradient is analytically given as

$$\frac{\partial f_{\mathrm{mp}}^{(\mathrm{k})}}{\partial x} = 1 \tag{37}$$

$$\frac{\partial f_{\mathrm{mp}}^{(\mathrm{k})}}{\partial y} = \frac{y}{R_{\mathrm{mp}}} \tag{38}$$

The magnetopause-normal direction $e_{\mathrm{mp}}^{(\mathrm{k})}$ is then obtained by applying Eqs. (37) and (38) to Eq. (26), which reads as

$$\begin{aligned} e_{\mathrm{mp}}^{(\mathrm{k})} &= \frac{\mathrm{sgn}}{\sqrt{(\partial_x f_{\mathrm{mp}}^{(\mathrm{k})})^2 + (\partial_y f_{\mathrm{mp}}^{(\mathrm{k})})^2}} \times \\ &\quad \left( \partial_x f_{\mathrm{mp}}^{(\mathrm{k})}\, e_{\mathrm{x}} + \partial_y f_{\mathrm{mp}}^{(\mathrm{k})}\, e_{\mathrm{y}} \right) \end{aligned} \tag{39}$$

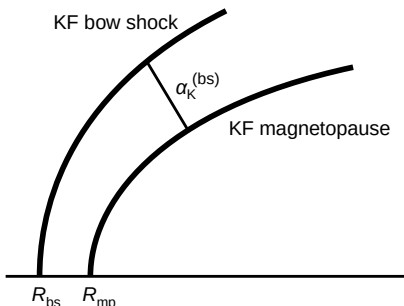

**Figure 7.** Computing the magnetosheath thickness in the KF model (Step 3). The same zenith angle as that in step 2 is used.


The thickness in the KF system $\alpha_{\mathrm{k}}^{(\mathrm{bs})}$ is determined by combining the bow shock shape (Eq. 31) with the position vector at the bow shock:

$$r_{\mathrm{bs}}^{(\mathrm{k})} = r_{\mathrm{mp}}^{(\mathrm{k})} + \alpha_{\mathrm{k}}^{(\mathrm{bs})} e_{\mathrm{mp}}^{(\mathrm{k})}. \tag{40}$$

Equation (31) becomes again a quadratic equation with respect to the thickness $\alpha_{\mathrm{k}}^{(\mathrm{bs})}$, and the solution reads:

$$\begin{aligned} \alpha_{\mathrm{k}}^{(\mathrm{bs})} &= \frac{1}{2b_{\mathrm{k}}\, (e_{\mathrm{mp},y}^{(\mathrm{k})})^2} \times \\ &\quad [-(e_{\mathrm{mp},x}^{(\mathrm{k})} + 2b_{\mathrm{k}}\, y_{\mathrm{mp}}\, e_{\mathrm{mp},y}^{(\mathrm{k})}) + d_{\alpha}^{(\mathrm{k})}] \end{aligned} \tag{41}$$

where the auxiliary variable $d_{\alpha}^{(\mathrm{k})}$ is defined as

$$\begin{aligned} d_{\alpha}^{(\mathrm{k})} &= [(e_{\mathrm{mp},x}^{(\mathrm{k})} + 2b_{\mathrm{k}}\, y_{\mathrm{mp}}\, e_{\mathrm{mp},y}^{(\mathrm{k})})^2 - \\ &\quad -4b_{\mathrm{k}}\, (e_{\mathrm{mp},y}^{(\mathrm{k})})^2 (x_{\mathrm{mp}} + b_{\mathrm{k}}\, y_{\mathrm{mp}}^2 - R_{\mathrm{bs}})]^{1/2}. \end{aligned} \tag{42}$$

## 4.6 Step 4: Mapping the position vector onto the KF system

In the fourth step, the mapping of the position vector is performed from the empirical magnetosheath onto the KF system (Fig. 8). Assumption is made such that the relative distance to the magnetopause along the magnetopause-normal direction is the same between the two systems. The distance from the magnetosheath position vector to the magnetopause along the magnetopause-normal direction in the KF system $\alpha_k$ is then determined by the relative distance in the empirical magnetosheath $\alpha_{emp}$, the thickness of the empirical magnetosheath $\alpha_{emp}^{bs}$, and magnetosheath thickness in the KF system $\alpha_k^{(bs)}$ as

$$\alpha_k = \alpha_{emp}\, \alpha_k^{(bs)}/\alpha_{emp}^{(bs)}. \tag{43}$$

The mapped position vector is then computed as

$$\boldsymbol{r}^{(k)} = \boldsymbol{r}_{mp}^{(k)} + \alpha_k\, \boldsymbol{e}_{mp}^{(k)}, \tag{44}$$

using the nearest magnetopause position $\boldsymbol{r}_{mp}^{(k)}$ (Eq. 35), the magnetosheath-to-magnetopause distance $\alpha_k$ (Eq. 43), and the magnetopause-normal direction $\boldsymbol{e}_{mp}^{(k)}$ (Eq. 39).

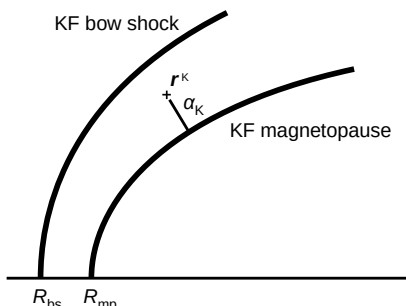

**Figure 8.** Mapping the position vector onto the KF magnetosheath model (Step 4).

## 4.7 Step 5: Evaluating the shell and connector variables

In the fifth step, the shell variable $v$ and the connector variable $u$ are computed from the mapped position vector $\boldsymbol{r}^{(k)}$ using Eqs. (1) and (2). respectively. The variables $v$ and $u$ are the same as the parabolic coordinates used in the KF potential with a focus at $x_0 = R_{mp}/2$. In our algorithm, the focus is explicitly given in Eqs. (1), (2), and (3). The azimuthal angle around the symmetry axis $\phi$ is treated in the same way as in the KF model.

Figure 10 compares the iso-contours of the shell $v$ and the connector $u$ represented in the KF system (left panel) and the empirical magnetosheath (right panel) for a bow shock stand-off distance of 12.8 $R_E$ (Génot et al., 2011), a bow shock flaring of 0.0223 $R_E^{-1}$, (Farris et al., 1991; Cairns et al., 1995), and a magnetopause stand-off distance 9.8 $R_E$ (Génot et al., 2011). The shell variable $v$ is characterized by the lines with the curvature center on the right side in the panel, and contains the

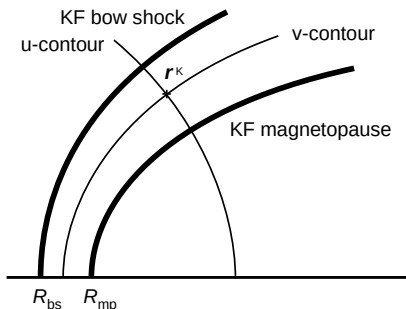

**Figure 9.** Evaluating the shell variable $v$ and the connector variable $u$ in the KF magnetosheath model (Step 5).

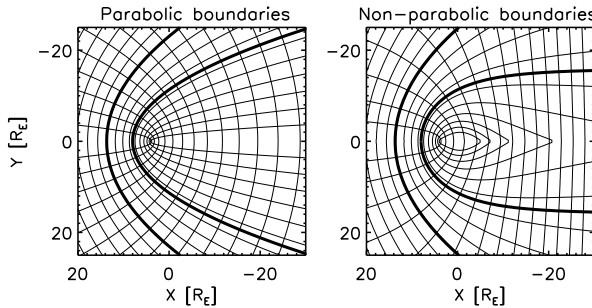

**Figure 10.** Iso-contour lines with $u = \text{const.}$ (center of curvature on the left side) and that with $v = \text{const.}$ (center of curvature on the right side) in the KF magnetosheath model (left panel) and the empirical magnetosheath model (right panel). The bow shock stand-off distance is 12.8 Earth radii and the magnetopause stand-off distance is 9.8 Earth radii.

parabolic bow shock (at $v = v_{\mathrm{bs}}$) and magnetopause (at $v = v_{\mathrm{mp}}$) marked by thick lines. The connector variable $u$ has the curvature center on the left side in the panel, and the iso-contour lines are orthogonal to the bow shock and magnetopause. The computation of the $u$ and $v$ variables and their gradient and curl is performed in Cartesian coordinates so that the connection represented by the Christoffel symbol vanish in the computation. Computation in the Cartesian domain is also beneficial to the

practical application because spacecraft trajectories are often represented in the Cartesian coordinates.

### 4.8   Step 6: Computing the potentials and stream function

The scalar potentials (velocity potential and magnetic potential) and the stream function are obtained from the shell $v$ and the connector $u$ using Eqs. (10), (13), and (15). The velocity potential (normalized to the upstream flow) is displayed in Fig. 11 left panel, and the stream function in right panel. The iso-contours of the velocity potential represent the lines of the same

flow velocity. The iso-contours of the stream function represent the streamlines in the magnetosheath. The flow is deflected around the nose of the magnetopause (the subsolar point) and the streamlines are tangential to the magnetopause. Due to the

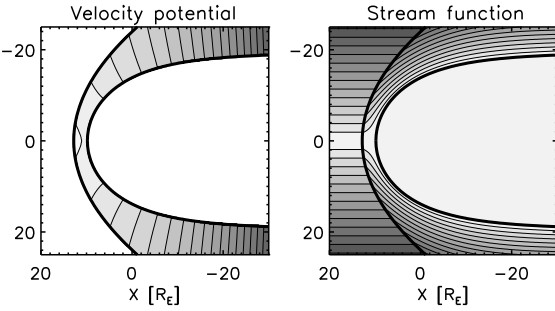

**Figure 11.** Velocity potential (left panel) and stream function (right panel) in the empirical magnetosheath domain obtained by mapping onto the shell variable $v$ and the connector variable $u$. Note that two different functions are plotted here for the same grid and mapping method. The color range is chosen for the visual demonstration from $5.5R_{\mathrm{E}}$ to $314R_{\mathrm{E}}$ (left panel) and from $0R_{\mathrm{E}}$ to $15R_{\mathrm{E}}$ (right panel).

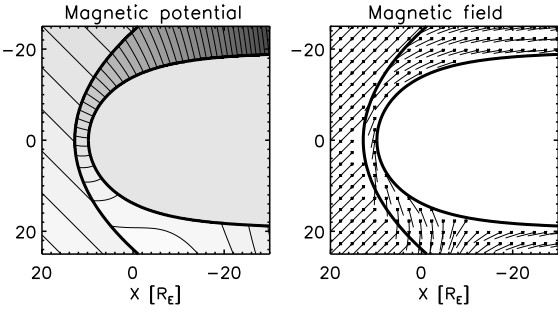

**Figure 12.** Magnetic potential for the upstream magnetic field with an angle of 135 degree to the x axis (45 degree to the upstream flow direction, (left panel) and sampled magnetic field vectors obtained by the negative gradient of the magnetic potential (right panel). The color range of the magnetic potential is from $-35R_{\mathrm{E}}$ to $348R_{\mathrm{E}}$.

grid orthogonality around the magnetopause, the streamlines are constructed as tangential to the magnetopause shape, which qualifies the magnetopause-normal mapping method as a useful tool for the magnetosheath model.

The magnetic potential and the derived magnetic field are displayed in Fig. 12. The magnetic potential and the magnetic field
(the gradient of the potential multiplied by the minus sign) depend on the upstream field. Fig. 12 shows an example with an upstream field angle of 135 degree to the x axis (i.e., 45 degree to the upstream flow direction). The magnetic field is computed using the central difference scheme. Near the boundaries (bow shock or magnetopause), the mesh resolution is enhanced so that the mesh points do not cross the boundary when computing with the central difference scheme. The upstream field is deflected on the positive y side (right panel, lower half plane), and is draping the magnetopause on the negative y side (right
panel, upper half plane).

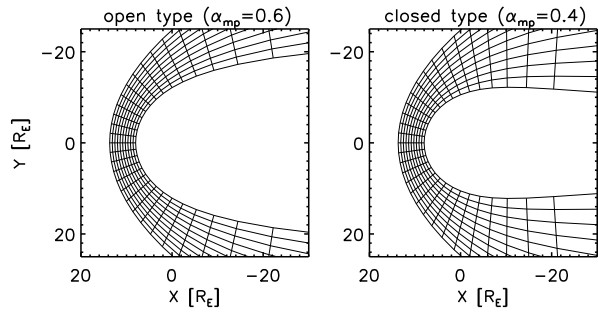

**Figure 13.** Mesh pattern applied to different values of the Shue exponent for an open-type magnetopause $\alpha_{\mathrm{mp}} = 0.6$ (left panel) and a closed-type magnetopause $\alpha_{\mathrm{mp}} = 0.4$ (right panel).

## 5    Discussions

### 5.1    Extension of the mapping approach

It is straightforward to extend our method to different shapes of the Shue magneotpause model. The following form of gradient can be used for a general value of the Shue exponent $\alpha_{\mathrm{mp}}$,

$$\frac{\partial f_{\mathrm{mp}}}{\partial x} = \cos\theta + \frac{2_{\mathrm{mp}}^{\alpha} R_{\mathrm{mp}} \alpha}{r} \left(1 + \cos\theta\right)^{-(\alpha_{\mathrm{mp}}+1)} \sin^2\theta \tag{45}$$

$$\frac{\partial f_{\mathrm{mp}}}{\partial y} = \sin\theta - \frac{2_{\mathrm{mp}}^{\alpha} R_{\mathrm{mp}} \alpha}{r} \left(1 + \cos\theta\right)^{-(\alpha_{\mathrm{mp}}+1)} \cos\theta \sin\theta, \tag{46}$$

where $f_{\mathrm{mp}}$ in Eqs. (45)–(46) is defined as

$$f_{\mathrm{mp}} = r - R_{\mathrm{mp}} \left(\frac{2}{1 + \cos\theta}\right)^{\alpha_{\mathrm{mp}}}. \tag{47}$$

Mesh pattern applied to different values of the Shue exponent for an open-type magnetopause $\alpha_{\mathrm{mp}} = 0.6$ (Fig. 13 left panel) and a closed-type magnetopause $\alpha_{\mathrm{mp}} = 0.4$ (Fig. 13 right panel).

Also, our method can be extended to a three-dimensional, non-axisymmetric geometry of magnetosheath (e.g., Dimmock and Nykyri, 2013). To achieve this goal, a suitable set of the variables needs to be found for the mapping: the shell variable, the connector variable, and the aximuthal angle around the subsolar axis (solar wind direction intersecting the planetary magnetic dipole). Namely, one needs to give a non-axisymmetric bow shock shape and a non-axisymmetric magnetopause shape, compute the magnetopause-normal direction and constructing grids in the magnetosheath, measure the distance to the magnetopause and the bow shock, scale and relate the distance to the KF model, and evaluate the scalar potentials through the $u$, $v$, and $\phi$ variables.

### 5.2    Other approaches

It is possible to obtain the steady-state magnetosheath potential in different ways.

– First, one may numerically solve the Laplace equation for a given set of boundary shapes (bow shock and magnetopause). Various numerical solvers are known for solving the Laplace equation such as the Jacobi method, the Gauss-Seidel method, and the successive over-relaxation (SOR) method. These Laplace solvers are numerically more expensive than the mapping method, but the computation in 3-D is feasible with the contemporary computational resources. On the other hand, the magnetosheath is not bounded, but extends in the tail direction. The challenge here is thus to construct a

properly bounded area for the Laplace equation.

– Second, one may expand the magnetosheath magnetic field in different orthogonal functions. The KF solution makes use of the Bessel functions (Kobel and Flückiger, 1994). For flexible magnetopause and bow shock boundary models, a magnetosheath magnetic field model is constructed by making use of Legendre polynomials (Romashets and Vandas, 2019).

– Third, one may introduce a suitable conformal mapping by limitinng the magnetosheath modeling to a complex plane (two-dimensional domain). The harmonic functions such as the KF solution are transformed from the parabolic magnetosheath shape into the non-axisymmetric mganetosheath shape. The problem here is that finding the conformal mapping is not an easy task because the magnetosheath is not a spatially-closed domain and one has to set the boundary in the magnetosheath to complete the domain bounded by the bow shock, the subsolar axis, and the magnetopause.

– Fourth, one may solve the Rankine-Hugoniot relations and track the streamline step-wise by refererring to the KF solution, as is done in Soucek and Escoubet (2012). This method computes the magnetosheath flow and magnetic field along the streamline, and the computing for the entire magnetosheath domain is numerically expensive.

## 6   Concluding remark

Our potential mapping method may be regarded as an updated version of the radial mapping method (Soucek and Escoubet, 2012) by retaining the orthogonality near the magnetopause in the flank to tail region and also by computing the field through the potential mapping. Velocity potential, stream function, and magnetic potential are evaluated in the empirical magnetosheath. The advantages of our methods are as follows.

1. The method makes extensive use of the exact solution of the Laplace equation (the Kobel-Flückiger potential and the Guicking stream function). The plasma flow and magnetic field can be determined semi-analytically in a wide range of zenith angle in the magnetosheath when the solar wind conditions and the boundary shapes are given.

2. The method is applicable to a non-parabolic shape of magnetosheath domain, opening the door to develop a tool to assist numerical simulations and spacecraft observations of not only the Earth but also the planetary magnetosheath domain.

3. The method is computationally inexpensive. In particular, if the shape of bow shock and magnetopause is analytically given, most of the computational steps in the potential mapping method have an analytic expression.

As stated in section 1, one ideally needs to find a conformal mapping from the KF magnetosheath model onto the empirical magnetosheath. While the conformal mapping is known both for the empirical bow shock and the empirical magnetopause, the conformal mapping of the entire magnetosheath domain still remains a challenge. There are two problems with this approach. First, the closing boundary (the $u$-contours) connecting between the bow shock and the magnetopause is not known, and moreover, the uniqueness of finding such a boundary is not guaranteed. Second, the gradients along $u$ are not the same between

the empirical bow shock and the empirical magnetopause such that a naive transfinite interpolation ends up with highly non-orthogonal grids in the magnetosheath.

Our method of computing the plasma flow and magnetic field should be compared against the observations and simulations. For example, THEMIS and ARTEMIS spacecraft (Angelopoulos, 2008) and MMS spacecraft (Burch et al., 2016) are providing a huge amount of data on both sides of the bow shock in the equatorial plane; Cluster spacecraft (Escoubet et al., 2001) are

collecting data in polar orbit; ACE spacecraft data (Stone et al., 1998) may be used as an upstream monitor; and Earth flyby data of planetary missions (such as Cassini, BepiColombo) cover the far-distance tail region. In reality, non-axisymmetric structure arises in the magnetosheath. There is no restriction regarding the choice of the magnetopause model. The magnetopause-normal direction needs to be computed either analytically using the gradient of the magnetopause function as $\nabla f_{\mathrm{mp}}$, or numerically for a user-defined magnetopause shape.

**Appendix: Planet-to-bow shock distance**

By introducing the zenith angle $\theta$ and inserting $x = r_{\mathrm{bs}} \cos\theta$ and $y = r_{\mathrm{bs}} \sin\theta$ in Eq. 16), we obtain the equation for the radial distance to the empirical bow shock:

$$b_{\mathrm{emp}}\, r_{\mathrm{bs}}^2 \sin^2\theta + r_{\mathrm{bs}} \cos\theta - R_{\mathrm{bs}} = 0. \tag{48}$$

Equation (48) can be algebraically solved, and we take the positive value of the solution as presented in Eq. (19).

*Code and data availability.*  No codes or data are used in this work.

*Author contributions.*  YN, ST, and DS developed the idea of potential mapping method, checked mathematics, and wrote the manuscript. YN prepared the figures. All authors listed have made a substantial, direct, and intellectual contribution to the work and approved it for publication.

*Competing interests.*  Conflict of Interest: The authors declare that the research was conducted in the absence of any commercial or financial

relationships that could be construed as a potential conflict of interest.

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
