# Peer review of "Scalar-potential mapping of the steady-state magnetosheath model"

_Annales Geophysicae, 2023_

## Author Comment (AC1)

**Referee 1**

1. *This manuscript describes a methodology for mapping magnetosheath locations relative to specific empirically-based models for the bow shock and magnetopause into an equivalent magnetosheath location with boundaries described by confocal paraboloids. Analytic solutions for plasma streamlines (and potential) and magnetic fields (and magnetic potential) can then be conformally mapped to a space bounded by more realistic boundaries.*

   **Reply (ref.01.01)**:

   - No, not exactly. We are transforming the magnetosheath scalar potential not by a conformal mapping but by a non-conformal (and non-orthogonal) mapping in this manuscript. It is of course ideal if the harmonic functions (given as the solution of Laplace equation) were transformed into an arbitrarily-bounded magnetosheath using the conformal mapping. After extensive theoretical research (both analytically and numerically), it became clear that the conformal mapping of magnetosheath cannot be constructed uniquely. The reason for this is that the magnetosheath is not properly bounded for solving the Laplace equation. The magnetosheath is bounded only in the radial direction from the planet (or normal to the magnetopause) by the bow shock and the magnetopause. there is no boundary along the streamline, and the conformal mapping (Cauchy-Riemann condition or orthogonality condition) is no longer unique. Nevertheless, the algorithm we develop in the manuscirpt is a useful approach, because one can utilize the analytic solutions and the algorithm can relatively easily be implemented to various boundary shapes (though we chose only one example), which is of great help for future planetary research (missions and simulations). We highlight the problem with the conformal mapping in section 1 (page 2, lines 32–39) and section 5 (page 16, lines 314–320).

2. *In general, this article does not represent a significant advancement. It reads more like an Appendix of a larger study, with the Appendix detailing a technique to map locations between confocal, parabolic boundaries and empirically-based boundaries. While this technique is similar to previous efforts as described by Soucek and Escoubet [2012], Trattner et al., JGR [2015], and others, there is no effort here to demonstrate that this particular mapping technique better matches observations than previous techniques.*

   **Reply (ref.01.02)**:

   - W accept the critique that the manuscript reads more like an appendix of thesis. This impression comes from the slight mismatch between the manuscript goal (tool or algorithm development) and the journal scope (such as scientific message). We aim to develop a numerical grid scheme for space science applicaitons. Numerical grid schemes are one of the favored discussion topics in fluid dynamics, computational physics, and informatics, but not so widely acknowledged in space science journals. See, for example, a grid generation using the conformal mapping by

Lin and Chandler-Wilde, J. Hydroinformatics, 2, 255–267, 2000 https://doi.org/10.2166/hydro.2000.0023. We nevertheless choose AnGeo for the dissemination of our study because the space science community should benefit the most from our algorithm development.

- The drawback with the radial mapping by Soucek and Escoubet (2012) is that the quality of mapping (distortion effect due to non-orthogonal grids) becomes quickly degraded in the flank to distant-tail region. Our mesh is robust against the distortion effect in the tail region. This point is elaborated in section 3 (pages 4–6) in the revised manuscript.

3. *Some of the references to empirical models of boundary shapes/sizes are inconsistent with the description provided here, or are examined under extremely specific solar wind conditions, or do not properly represent the knowledge of the physical boundaries far down the flank. Specifically,*

   (a) *The Farris et al., JGR [1991] empirical bow shock model is not a paraboloid model. It is an ellipsoid model (eccentricity of 0.81), describing the bow shock on the dayside. It is not a proper representation of the far flank bow shock.*

   (b) *The Cairns et al., JGR [1995] paraboloid bow shock model also does not properly represent the far flank bow shock. The distant bow shock shape approaches that of a hyperboloid.*

   (c) *The authors have selected a very specific exponent for the Shue et al., JGR [1997] model (alpha = 0.5) in an effort to show that the analytic model is 'simple'. The solar wind conditions for which this exponent is applicable (from the Shue et al. mode) is not often encountered (IMF Bz ¿ +8 nT, with specific values of solar wind dynamic pressure in order that alpha = 0.5).*

   **Reply (ref.01.03)**:
   - True and agreed. We added the referee's comments (page 8, lines 160–164 and lines 170–172).

4. *Additional references to analytic models of the magnetosheath magnetic field (using expansions in Legendre polynomials) that make use of flexible magnetopause and bow shock boundary models (e.g., Romashets and Vandas, JGR, [2019]) should be provided and discussed.*

   **Reply (ref.01.04)**:
   - Oh, thank you very much for introducing us this excellent paper! Yes, we cited the paper (page 16, lines 333–335).

5. *Although the claim is made in the manuscript that this technique can be applied to arbitrary boundary model shapes, it is not demonstrated that under general*

*circumstances, the equations can be written in a closed form.*

**Reply (ref.01.05)**:

- Critique is well taken. We changed "arbitrary" into "non-parabolic" as we applied only one example (page 15, line 310).

6. *The technique described relies on determining the (straight line, or minimum) distance from a given point within the magnetosheath to the magnetopause. This is along the normal direction from the magnetopause. However, Lines 109–110 state that the task is to find the shell variable 'v' and the connector variable 'u' in the empirical magnetosheath. However, while the connector variable 'u' of the empirical magnetosheath is normal to the magnetopause surface, it is not a straight line through the magnetosheath? and doesn't represent the minimum distance from the given point to the magnetopause. In other words, the distance from the magnetopause extends over a (narrow) range of connector variable 'u' values.*

**Reply (ref.01.06)**:

- The confusion comes from the difference between the grids we used in the mapping (shown in figure 3, page 6, in the revised manuscript) and the u-v contours we showed in the original manuscript. We span the magnetopause-normal grids both in the KF model and in the empirical model, and here we see straight lines extending to the magnetopause.

7. *The rationale for the methodology described is confusing. For most implementations, the solar wind parameters are known, and the corresponding parameters at a given point within the magnetosheath are desired. However, the methodology here is to start with known parameters at a given place within the magnetosheath (relative to empirical models), conformal map to a location relative to the KF paraboloid boundaries, calculate the 'u' and 'v' values and determine the B-field, streamline, and potentials. The solar wind drivers appear to be missing. It appears that part of this strategy is based on the Toepfer et al. [2022] motivation; but a clear description for the order of steps for this technique is missing.*

**Reply (ref.01.07)**:

- Yes, in the case of space plasma missions orbiting the Earth; But no, in the case of planetary mission (in which even the availability of plasma data is still chanllenging due to the mass, power budget, and telemetry budget). The advantage with the manuscript is that one can compute the magnetosheath potential for various scenarios of upstream conditions without extensive numerical efforts, assuming that the boundaries (bow shock and magnetopause) are well parametrized to the solar wind condition.

8. *Several of the equations presented are incorrect. For example, Eq.5 is infinite everywhere, due to the denominator. How are Eqs. 20 and 23 are used to derive Eq.24? Why do the units not match for the terms within Eq.39? How do Eqs.31-32 lead to Eq.33 when ymp=0?*

   **Reply (ref.01.08)**:

   - Equation (5). Corrected (page 4, Eq. 13). Thank you!
   - It is straightforward to derive Eq. (24) from Eq. (23), but Equation (24) offers an alternative approach to find the minimum distance to the brute-force method, but the manuscript can read even withtout this equation. Equation (24) was deleted in the revised manuscript.
   - Equation (39). "2" should read "x" (page 11, Eq. 36). Thank you!

[revised manuscript text omitted]

---

## Author Comment (AC2)

**Referee 2**

1. *General comments*

   *This manuscript proposes a method for generalizing the mapping of flow lines and magnetic field in the magnetosheath. The proposed method aims to be computationally inexpensive and generalizable, which is desirable for many applications including statistical studies. However, a number of critical issues need to be addressed if the manuscript is to be considered for publication. The results presented are not general enough to be of use for actual applications; thus, the manuscript represents no (or very minor) scientific advancement. The novelty of the study needs to be explained, in particular by a more focused comparison with previous works. Furthermore, the method should be presented more clearly; it is currently difficult to tell if the proposed method is incorrect or if the presentation is unclear and contains too many mistakes (typos and inconsistencies). Please find below my detailed comments and suggestions for improvement.*

   **Reply (ref.02.01)**:

   - Thank you for suggestion that the novelty should be clear in the manuscript. We compare the grid pattern between the radial mapping by Soucek and Escoubet (2012) (figure 1, page 5) and our magnetopause-normal mapping (figure 3, page 6). The radial mapping has the problem of artificial converging flow pattern in the flank region (see figure 2 right panel, page 5). This is due to the strong grid non-orthogonality. Our method can properly transform the scalar-potential and there is no artificial converging flow pattern (figure 11, page 15).

2. *Specific comments*

   *Concerning the whole manuscript*

   *Coordinate system: The authors introduce a new, non-parabolic coordinate system (u,v,φ) in which the potentials for the velocity and magnetic field are to be expressed. However, it is not mentioned whether the new coordinate system is orthogonal; in fact, from Fig. 3 it appears that the grid in the right panel is not. The parabolic coordinate system (used in KF94) is orthogonal by construction, and the gradient and Laplace equation in this system are defined and given explicitly (Eq. (8)-(9) in the KF94 paper). In this manuscript, however, the potentials are obtained from the shell and connector variables v and u in the new coordinate system (according to line 274-285 and 289). The authors do not define the gradient and curl in this system and thus the evaluation of Eq. (1) is not defined in the manuscript.*

   **Reply (ref.02.02)**:

   - The coordinates with $u$ and $v$ are orghotonal around the magnetopause, and the grid non-orthogonality is suppressed in our method (section 3.3, page 5 and figure 3, page 6; ). To make it fully orthogonal (curvilinear

and locally orthogonal grids) one needs to find a conformal mapping. Mathematical studies conclude that there is no unique conformal mapping to the magnetosheath problem because the magnetosheath is bounded only by the magnetopause and the bow shock in the radial direction but not along the streamlines.

- The azimuthal coordinate $\phi$ is still orthogonal to the $u$ and $v$ coordinates (page 5, lines 111–112).

- The computation of the $u$ and $v$ variables and their gradient and curl is performed in the Cartesian so that the connection represented by the Christoffel symbol vanish in the computation. Computation in the Cartesian domain is also beneficial to the practical application because spacecraft trajectories are often represented in the Cartesian coordinates (page 14, lines 283–286).

3. *Generalization of the method: In section 3, the mapping algorithm is reduced to an axisymmetric geometry and the calculations are made specifically for the Farris (1991) bow shock and Shue (1997) models. Yet, it is stated in sections 1 and 4 that the method is easy to generalize to an arbitrary magnetopause shape. Would it be possible to express the derivations in more general terms? I suggest giving expressions on a form which allows for non-axisymmetric geometries and aberrated GSE coordinate systems (see for example the asymmetric magnetosheath thickness and aberrated x axis in https://doi.org/10.1002/jgra.50465), or otherwise indicate to the reader which modifications and transformations are necessary to generalize the method. Are there restrictions on the choice of magnetopause model? Does the method require an analytic expression for the magnetopause?*

   **Reply (ref.02.03)**:

   - Non-axisymmetric case. It is possible to obtain the steady-state magnetosheath potential in a more general sense without referring to the KF94 solution. For example, for a non-axisymmetric geometry of magnetosheath (e.g., Dimmock and Nykyri, 2013), one needs to sove the Laplace equation for a given set of boundaries (bow shock and magnetopause). Various numerical solvers are known for solving the Laplace equation such as the Jacobi method, the Gauss-Seidel method, and the successive over-relaxation (SOR) method. These Laplace solvers are numerically more expensive than the mapping method, but the computation in 3-D is feasible with the contemporary computational resources (page 16, lines 327–332).

   - Magnewtopause model. There is no restriction regarding the choice of the magnetopause model. The magnetopause-normal direction needs to be computed either analytically using the gradient of the magnetopause function as $\nabla f_{\mathrm{mp}}$, or numerically for a user-defined magnetopause shape (page 16, lines 335–337).

4. *Figures 1 and 2 show examples of results of the mapping, but not in which way the proposed method is better than (or even different from) the method*

*by Soucek (2012). They are also introduced very early in the manuscript, before the potential functions or the mapping procedure have been described, and do not contain much relevant information. I suggest replacing them with figures showing the steps of the mapping procedure and/or a comparison with the Soucek (2012) method.*

**Reply (ref.02.04)**:

- Figure 1 and figure 3 compare directly the quality of grid pattern between the method by Soucek and Escoubet (2012) and ours. The artificially introduced converging flow pattern in the radial mapping method is shown in figure 2 (page 6).
- Agreed. Stepwise figures are shown in the revision (figures 5–9).

5. *Title*

   *The phrase "Potential mapping method" can be interpreted as "Possible mapping method". Perhaps the title could be rephrased to avoid misunderstanding.*

   **Reply (ref.02.05)**:

   - Agreed. We change the title into "Scalar-potential mapping of the steady-state magnetosheath model" to avoid confusion (title field, page 1).

6. *Abstract*

   *The abstract lacks a motivation for the study (a short background and a science question to be answered). It is very technical, especially the second sentence, and difficult to follow before reading the manuscript.*

   **Reply (ref.02.06)**:

   - Agreed. The abstract was rewritten by including the short background (page 1, lines 1–3) and the science question (page 1, lines 3–4).

7. *Section 1*

   *This section needs to be more concise and explain the novelty of the proposed method. In particular, the advantages compared to Soucek an Escoubet (2012) need to be clearly explained. Furthermore, since this study is very similar to the present work, it would be informative if the manuscript contained a comparison between the two methods in cases where the difference between them is the most important, together with an explanation as to why the proposed method is advantageous.*

   **Reply (ref.02.07)**:

   - Agreed. We have cut the introduction text and added paragraph addressing the question (page 2, lines 40–44).

8. *The introduction should be restructured. For example, lines 36-38 "While the radial mapping [...]") say the same thing as lines 52-53 ("While the radial mapping [...]"). The "gap" that this study fills is also stated twice (line 25-26 and line 48-49).*

   **Reply (ref.02.08)**:

   - Agreed. We shortened the introduction (section 1, pages 1–2), and restructured the manuscript with the review of KF94 model (section 2, pages 2–4) and discussion about different mapping methods (section 3, pages 4–6).

9. *Line 52-53: "While the radial mapping is nearly boundary-fitted on the dayside, the orthogonality of mapping degrades in the flank to tail region.": this sentence is essential as it mentions the difference between the proposed method and the previous one. However, the term "orthogonality of mapping" should be better explained and perhaps illustrated in a figure. Currently, it is not clear what "orthogonality" refers to.*

   **Reply (ref.02.09)**:

   - Agreed. The orthogonality (or non-orthogonality) of grids is graphically displayed in figure 1 (page 5) and figure 3 (page 6).

10. *Section 2*

    *The title should be more informative, for example indicating that the section reviews theory from previous works.*

    **Reply (ref.02.10)**:

    - New section headers are:
        - Sec. 1 – Introduction
        - Sec. 2 – Revisiting the magnetosheath scalar potential
            * 2.1    Parabolic coordinates
            * 2.2    Velocity potential
            * 2.3    Stream function
            * 2.4    Magnetic scalar potential
        - Sec. 3 – Mapping method comparison
            * 3.1    Mapping problem
            * 3.2    Radial direction as reference
            * 3.3    Magnetopause-normal direction as reference
        -
            * 4.1    Overview of the procedure
            * 4.2    Setup
            * 4.3    Step 1: Measuring the distance to magnetopause
            * 4.4     Step 2: Computing the thickness of empirical magnetosheath

11. *This section reviews previous results from KF94. It should begin with an introductory sentence explaining what the section contains. It was not entirely clear when the review of previous work ends and the new work starts. Also, the level of detail in the section seems a bit unnecessary. Would the authors perhaps consider referring to the works by Kobel and Flückiger (1994) and Guicking et al. (2012), instead of writing out all the equations in the main text? Alternatively, detailed equations could be placed in an appendix to improve the flow in the text.*

**Reply (ref.02.11)**:

- We added a short paragraph about the section organization in the paper at the end of section 1 (page 1, lines 45–47).
- It is better for the benefit to the readers if the manuscript contains all the necessary information (equations) in a coherent way, rather than simply citing the original referencces. The reason for this is that the coordinate system used by Kobel and Flückiger (1994) has the co-focal point as the origin. In our method, the bow shock and the magnetopause do not share the same focal point.

12. *Section 3*

    *Section 3.1: This section contains confusing terminology and probably some typos. These include:*

    - *Line 130: "magnetosheath-to-magnetopause distance": the magnetosheath is a region bounded by the magnetopause, so this phrase does not make sense. Instead of magnetosheath, consider writing "position vector r" (since $\alpha emp$ is the distance from r to the magnetopause).*
    - *"magnetosheath-normal direction": → "magnetopause-normal direction" (line 136 and Figure 4 caption).*
    - *"magnetopause thickness" → "magnetosheath thickness" (line 137, Figure 4 caption, line 250, possibly more places in the text).*

**Reply (ref.02.12)**:

- Corrected into "computing the distance to the magnetopause" (page 7, line 140)

- Corrected into "the distance from the planet to the bow shock $r_{\text{bs}}$, the distance from the planet to the magnetopause $r_{\text{mp}}$, the relative distance to the magnetopause $\alpha_{\text{emp}}$" (page 7, lines 146–147).

- Corrected into "the magnetosheath thickness" (page 7, line 147–148).

13. *Line 138-139: are these the only input parameters, or should the magnetopause and bow shock shapes also be regarded as input?*

    **Reply (ref.02.13)**:

    - Yes, assuming that the shape of bow shock and that of magnetopause are already known. We changed the sentence into "The position vector $\boldsymbol{r}$, the bow shock stand-off distance $R_{\text{bs}}$, the bow shock shape, the magnetopause stand-off distance $R_{\text{mp}}$, and the magnetopause shape are assumed to be known in our mapping" (page 7, lines 148–149).

14. *Also, the reason for defining a unit vector orthogonal to the magnetopause was not clear when reading the manuscript for the first (or second) time. The motivation for this choice should be emphasized in this section and better explained in the introduction.*

    **Reply (ref.02.14)**:

    - The need for the magnetopause-normal direction is elaboted in section 3 (pages4–6). The grids need to be as orthogonal as possible particularly around the magneopause.

15. *Section 3.2-3.5: see above comments about generalization of the method. Equations (18)-(20) and (26)-(28) are specific to the Shue and Farris models; it should be clarified that they do not describe a generalized method.*

    **Reply (ref.02.15)**:

    - Equations (18)–(20) in the original manuscript are packed into a subsection "Setup" (section 4.2, pages 7–8). An explanation was added at the end of section 4.2 that one needs to compute the radial distance from the planet to the bow shock or magnetopause as a function of the zenith angle when using different shapes (page 8, lines 177–179).

    - Steps 1 and 2 in the original manuscript are compressed into section 4.2 in the revised manuscript. The procedure begins with Step 3 in the original manuscript, and is introduced as Step 1 in the revision (page 8, section 4.3).

16. *Section 3.6: The derivations are very similar to what has already been done in section 3.2-3.5. For the sake of getting a better flow in the text, perhaps it would be possible to reduce the number of equations, or make an appendix with the details?*

**Reply (ref.02.16)**:

- We thoroughly checked the logical flow in the text. We moved Eq. (17) in the original manuscript onto Eq. (47) in appexndix in the revised manuscript. We corrected Eqs. (13) and (36) in the revised manuscript. We deleted Eqs. (24) and (25) in the original manuscript. All the other equations are necessary and should appear as is.

17. *Section 3.8: Line 270-272: What is meant by the sentence "The mesh pattern [...]" Why is this important?*

**Reply (ref.02.17)**:

- The sentence was deleted. It becomes important when discussing the conformal mapping of magnetosheath coordinates, but we conclude that it is beyond the scope of the manuscript.

18. *Section 4*

    *This section needs to be reworked when the above comments have been taken into account.*

**Reply (ref.02.18)**:

- We went through the conclusion section after revising the main text part. We changed "arbitrary shape" into "non-parabolic shape" (page 15, line 310) and also added more discussion on page 16, lines 327–337.

19. *Line 292: "wider range" – compared to what? (This also appears in the abstract.)*

**Reply (ref.02.19)**:

- Corrected into "wide range" (page 1, line 12 and page 15, line 308).

20. *Line 294: "The method is applicable to an arbitrary shape of magnetosheath domain": in its current state, the method is specific to the Shue and Farris models, and thus this sentence is too strong (see above comments about generalization of the method).*

**Reply (ref.02.20)**:

- Changed into "non-parabolic shape" (page 15, line 310).

21. *Line 302-304: Here the authors mention non-orthogonality – is the coordinate system (u,v,φ) orthogonal in the magnetosheath?*

   **Reply (ref.02.21)**:

   - No, not orthogonal between u and v any more due to the non-uniqueness or non-existence of conformal mapping in the magnetosheath. But our method restores the orthogonality near the magnetopause.

22. *Line 309-311: "In reality, non-axisymmetric [...]" – this should be expanded on or incorporated in other parts of the manuscript.*

   **Reply (ref.02.22)**:

   - Done (page 16, lines 327–332).

23. *Technical corrections*

   - *Line 4: "solution of Laplace equation" → "solution of the Laplace equation"*
   - *Line 51: "magnetopause. But" → "magnetopause, but"*
   - *Line 53: "orthogonality of mapping" → "orthogonality of the mapping"*
   - *Line 63: "flow velocity is U is" → "flow velocity U is?'*
   - *Line 164: "[...], the cylindrical distance" → "[...], and the cylindrical distance"*
   - *Line 207: "such that relative distance" → "such that the relative distance"*
   - *Line 277: "streamline" → "streamlines"*
   - *Line 278: "nose of magnetopause" → "nose of the magnetopause"*
   - *Line 293: "solar wind condition" → "solar wind conditions"*

   **Reply (ref.02.23)**:

   - "solutions" (page 1, line 8).
   - "magnetopause. But" was deleted.
   - "orthogonality of mapping" was deleted.
   - "flow velocity $U$ is" (page 3, line 68).
   - "and the cylindrical distance" (page 8, line 177).
   - "such that the relative distance" (page 10, line 212–213).
   - "streamlines" (page 14, line 291).
   - "nose of the magneopause" (page 14, line 292).
   - "solar wind conditions" (page 15, line 309).

[revised manuscript text omitted]

$$y_{\text{k}} = \boldsymbol{r}^{(\text{k})} \cdot \boldsymbol{e}_y \tag{6}$$

$$z_{\text{k}} = \boldsymbol{r}^{(\text{k})} \cdot \boldsymbol{e}_z. \tag{7}$$

To complete the variable set for computing the potentials and the stream function, the azimuthal angle $\phi$ is introduced as

$$\phi = \text{atan}(z_{\text{k}}/y_{\text{k}}). \tag{8}$$

**2.2 Velocity potential**

In the frame of potential field theory, the flow velocity $\boldsymbol{U}$ is obtained from the velocity potential (scalar potential) $\Phi^{(\text{vel})}$ as

$$\boldsymbol{U} = -\nabla \Phi^{(\text{vel})} = -\nabla \times \left( \Psi \boldsymbol{e}_\phi \right). \tag{9}$$

The symbol $\boldsymbol{e}_\phi$ is the unit vector in the azimuthal directions around the symmetry axis (Sun-to-planet direction). Kobel and
Flückiger (1994) and Guicking et al. (2012) obtained the analytic expression of the velocity potential $\Phi^{(\text{vel})}$ using the shell
variable $v$ (iso-contour lines enveloping the magnetosphere) and the connector variable $u$ (iso-contour lines connecting from
the bow shock to the magnetopause).

$$\Phi^{(\text{vel})} = -U_x \left( \frac{v_{\text{mp}}^2 v_{\text{bs}}^2}{v_{\text{bs}}^2 - v_{\text{mp}}^2} \right) \left( \frac{u^2 - v^2}{2v_{\text{bs}}^2} + \ln v \right) -$$
$$\frac{1}{2} U_x \left( u^2 - v^2 \right) + \Phi_0^{(\text{vel})}, \tag{10}$$

where $U_x$ is the upstream flow velocity, $v_{\text{mp}}$ the shell variable at the magnetopause, $v_{\text{bs}}$ the shell variable at the bow shock, $v$
the shell variable, $u$ the connector variable, and $\Phi_0^{(\text{vel})}$ a free parameter (integration constant) which is set to zero without loss
of generality. The boundary shell values $v_{\text{mp}}$ and $v_{\text{bs}}$ contain the information on the stand-off distances ($R_{\text{mp}}$ and $R_{\text{bs}}$) in the
subsolar region, and are defined by Kobel and Flückiger (1994) as

$$v_{\text{mp}} = \sqrt{R_{\text{mp}}} \tag{11}$$

[revised manuscript text omitted]

---

## Referee Report (RR1)

**Referee comments**

**Scalar-potential mapping of the steady-state magnetosheath model**

Round 2

**General comments**

The changes made by the authors improved the impression of the manuscript. Specifically, more focus is put on achieving a reasonable grid in the flank region rather than the ability to generalize the method, which is in line with the outcome of the study.

The added figures improved the clarity, especially Fig. 1-3. The updated section headers are also appreciated as they highlight the difference between previous works and the present study.

To represent a significant scientific advancement, the manuscript would need to be extended by benchmarking with real spacecraft data and comparing to the performance of previous methods (e.g. Soucek and Escoubet (2012)). Having said that, I will continue with more specific comments regarding the content of the current manuscript.

Still concerns regarding the generality of the mapping procedure and the presentation of the method.

**Specific comments**

The generalizability of the mapping procedure remains a bit unclear. The phrase "arbitrary shape" has been changed to "non-parabolic shape", but does it also need to be axisymmetric? The discussion reads:

line 326:
*"Our method has the possibility to be extended to three-dimensional, non-axisymmetric modeling by the use of magnetopause normal mapping. It is possible to obtain the steady-state magnetosheath potential in a more general sense without referring to the KF94 solution. [...] Various numerical solvers are known for solving the Laplace equation such as the Jacobi method, the Gauss-Seidel method, and the successive over-relaxation (SOR) method. These Laplace solvers are numerically more expensive than the mapping method, but the computation in 3-D is feasible with the contemporary computational resources."*

Here, it seems like the Laplace equation needs to be numerically solved for a 3D non-axisymmetric magnetosheath. But I thought your method was to use the analytic expressions from the KF solution and map them onto a magnetosheath with new boundaries. Is this not possible in the non-axisymmetric case? If so, this is quite a crude restriction which should be noted (perhaps in the introduction and/or around line 155).

Staying on the topic of the generality of the method, the following sentence is a bit strange:

line 170:
*"We use a specific exponent for the Shue model (with an alpha exponent of 0.5) in an effort to show that the analytic model is 'simple'. The solar wind conditions for which this exponent is applicable is not often encountered"*

This is a direct response to a previous referee comment. The impression is that you are only showing that the model is simple in a special case which is rarely encountered. With this result, you cannot claim that the general method is 'simple'. Thus, this sentence weakens your argument that the method is simple and/or computationally inexpensive. To improve credibility, would it be possible to give the results with a general alpha exponent?

The methods section still seems unnecessarily lengthy (compared to the scientific contribution of the study), since the same set of equations are repeated twice with only some changes in the notation. However, if the authors after thorough consideration regard all details as necessary, it can be included as-is.

I have a number of suggestions regarding the figures which might give them a more solid impression:

The figure titles are inconsistent – for example, in Fig. 11 the titles describe which functions are plotted and in Fig. 2 the titles refer to the grid and mapping method. The point of this study is that Fig. 11 (left panel) is different from Fig. 2 (right panel), so the 'structure' of the figures should be similar and Fig. 2 should be clearly referred to when discussing Fig. 11.

Instead of referring to the figure panels as left/right, why not introduce subfigures (e.g. Fig 2a)? I also suggest to add colorbars so that absolute numbers can be compared between the results of the different methods. In addition, the captions could probably be more informative.

It would be nice to have figures that should be compared with eachother side by side (e.g. Fig. 2 (left panel) vs Fig. 2 (right panel) vs Fig. 11 (left panel)), but I understand that this might not be reconcileable with the order in which they are referred to in the text. However, as stated above, there can be more references to the figures (e.g. Fig. 2 vs Fig. 11) when making comparisons.

Maybe combine Fig 1 and 3 to facilitate the comparison (keep all plots but make 2x2 subfigures).

On line 123, the reader might ask: You say that Soucek et al (2012) were able to avoid the problem, so why is your orthogonality needed?

**Technical corrections**

Text

- u and v are introduced on line 50 but defined/explained on line 72-73. Consider defining them where they are introduced.
- The stream function should be mentioned closer to Eq (9).
- Line 108-110 and line 116-118 are almost the same sentence, a bit repetitive.
- line 323-324: references are in the wrong format (parentheses).

Equations
- Eq (39): Parentheses in the denominator that should not be there.
- Eq (41): Are $e_{mp,x}$ and $e_{mp,y}$ the x and y components of $e_{mp}^{(k)}$? If so, they should have the superscript (k).

---

## Referee Report (RR2)

**Review of article Scalar-potential mapping of the steady-state magnetosheath model by Yasuhito Narita, Simon Toepfer, and Daniel Schmid**

I received the manuscript after it has gone through two iterations of the review and I believe the article is already in a good shape and describes a magnetosheath flow and magnetic field model which is relatively easy to implement and consistent with a range of empirical shock and magnetopause models. The mapping method is to a large extent analogous to earlier works, mapping the empirical magnetosheath geometry to an analytical model by Kobel and Fluckiger, but offers a likely improvement in model accuracy at the magnetosheath flanks and also allows an easy calculation of flow velocity and field geometry from the mapped potentials.

From this point of view, the article presents an improved method addressing a problem to which no standard solution currently exists, and thus shows potential for application in future studies.

I agree with one of the reviewers that it is hard to asses how useful the model is before it is compared to experimental data, but nevertheless, the article can be considered suitable for publication even in its present form, where only the method is presented, and the model can be tested by future studies, if considered useful. However, as a paper presenting a computational method, publishing the code along with the paper would be of a great benefit.

I have the following technical comments, which are mostly minor. If these are addressed, I believe the article can be published.

1) **Appendix** – the appendix is so short that I do not see a reason not to include the few lines in the text of the article. It would improve readability.

2) **Line 380, Code and data availability:** I believe the potential for future use of the model would be greatly enhanced by publishing the code for implementation of the potential mapping along with the article. This would allow others to build on this work more easily and to test it against spacecraft data.

The authors say "No codes or data are used in this work" which does not seem correct. The figures in the paper were certainly produced by a computer code implementing the mapping and in particular the code used to generate Figures 10, 11 and 12 would be a worthy and I would say almost mandatory attachment to this work.

---

## Author Response (AR2)

**Reply to the referee comments**

Manuscript ID: ANGEO-2023-2
Scalar-potential mapping of the steady-state magnetosheath model

Y. Narita

I thank the both referees for their careful reading and thoughtful comments. Reply to each comment is given here.
* * *
**Referee 1**

1. *This manuscript describes a methodology for mapping magnetosheath locations relative to specific empirically-based models for the bow magnetosheath location with boundaries described by confocal paraboloids. Analytic solutions for plasma streamlines (and potential) and mapped to a space bounded by more realistic boundaries. The authors have made significant efforts to provide more details and description of the techniques used, along with their advantages and to this referee's comments, concerns, and questions. There remain several minor issues with English usage, and some recommended edits, then this manuscript will be considered by this referee to be publishable in Annales Geophysicae.*

   *Line 1 (Abstract): 'applications in studying' → 'applications for studying'*

   **Reply (ref.01.01)**:

   - Done. Page 1, line 1.

2. *Line 2 (Abstract): 'the planetary magnetospheres In particular,' → 'planetary magnetospheres. In particular,'*

   **Reply (ref.01.02)**:

   - Done. Page 1, line 1.

3. *Lines 7-8 (Abstract): 'a set of the shell variable and the connector variable' → 'a set of variables (shell variable and connector variable)'*

   **Reply (ref.01.03)**:

   - Done. Page 1, lines 7–8.

4. *Line 9 (Abstract): 'zenith angle in the magnetosheath' → 'zenith angle within the magnetosheath relative to the magnetopause'*

   **Reply (ref.01.04)**:

   - "the zenith angle within the magnetosheath with respect to the direction to the Sun" on page 1, line 9.

5. *Line 13 (Abstract): 'by using the analytic expression' → ', using analytic expressions'*

   **Reply (ref.01.05)**:

   - Done. Page 1, lines 13–14.

6. *Line 14 (Abstract): 'is applicable' → 'and is applicable'*

   **Reply (ref.01.06)**:

   - Done. Page 1, line 14.

7. *Line 17: treated as ignored' → 'ignored'*

   **Reply (ref.01.07)**:

   - Done. Page 1, line 17.

8. *Line 25: 'on the assumption' → 'using the assumption'*

   **Reply (ref.01.08)**:

   - Done. Page 1, line 25.

9. *Line 28: 'solution; the' → 'solution. The' (it's better to break up this very long sentence)*

   **Reply (ref.01.09)**:

   - Done. Page 2, line 28.

10. *Line 33: 'an non-parabolic shape of empirical magnetosheath' → 'a non-parabolic empirical magnetosheath shape'*

    **Reply (ref.01.10)**:

    - Done. Page 2, lines 33.

11. *Line 70: directions → direction*

    **Reply (ref.01.11)**:

    - Done. Page 3, line 72.

12. *Lines 110-111: 'makes a difference to the radial mapping in using' → 'differs from the radial mapping by using'*

    **Reply (ref.01.12)**:

    - Done. Page 5, lines 111–112.

13. *Line 126: 'makes a difference' → 'differs'*

    **Reply (ref.01.13)**:

- Done. Page 5, line 135.

14. *Line 136: 'This is achieved on' → 'This is based on'*

    **Reply (ref.01.13)**:

    - Done. Page 6, line 145.

15. *Lines 137-138: 'is the same when normalized to the magnetosheath thickness (...)' → 'when normalized to the magnetosheath thickness (...) remains constant'*

    **Reply (ref.01.15)**:

    - "when scaled to the magnetosheath thickness (defined as the distance from the magnetopause to the bow shock along the magnetopause-normal direction) remains constant."

16. *Line 146: 'of the nearest magnetopause' → 'associated with the minimum distance to the magnetopause'*

    **Reply (ref.01.16)**:

    - Done. Page 7, line 155.

17. *Fig.4 caption: 'zenith angle of the nearest magnetopause' → 'zenith angle along the direction nearest to the magnetopause'*

    **Reply (ref.01.17)**:

    - Done. Page 7, figure 4 caption, line 1.

18. *Line 182: 'direction such as' → 'direction as'*

    **Reply (ref.01.18)**:

    - Done. Page 9, line 191.

19. *Line 193: 'Having the nearest magnetopause at a distance' → 'Using the minimum distance to the magnetopause'*

    **Reply (ref.01.19)**:

    - Done. Page 9, line 202.

20. *Line 221: 'simplify given as' → 'simply given by:'*

    **Reply (ref.01.20)**:

    - Done. Page 11, line 233.

21. *Line 226: 'Equations (28)' → 'Equation (28)'*

    **Reply (ref.01.21)**:

    - Done. Page 11, line 235.

22. *Line 226-227: 'set separately to avoid the numerical digergence problem.'* → *'set separately.'*

   **Reply (ref.01.22)**:

   - Done. Page 11, line 235.

23. *Line 284: 'in the Cartesian'* → *'in Cartesian coordinates'*

   **Reply (ref.01.22)**:

   - Done. Page 14, line 293.

24. *Line 316: 'problems on this.'* → *'problems with this approach.'*

   **Reply (ref.01.22)**:

   - Done. Page 18, line 362.

25. *Lines 338-339: 'Additional references to analytic models of the magnetosheath magnetic field (using expansions in Legendre polynomials) that make use of flexible magnetopause and bow shock boundary models (e.g., Romashets and Vandas, JGR, [2019])' This sentence is essentially repeated a few lines above, and should be removed*

   **Reply (ref.01.25)**:

   - Deleted.
* * *
**Referee 2**

1. *General comments*

   *The changes made by the authors improved the impression of the manuscript. Specifically, more focus is put on achieving a reasonable grid in the flank region rather than the ability to generalize the method, which is in line with the outcome of the study. The added figures improved the clarity, especially Fig. 1-3. The updated section headers are also appreciated as they highlight the difference between previous works and the present study. To represent a significant scientific advancement, the manuscript would need to be extended by benchmarking with real spacecraft data and comparing it to the performance of previous methods (e.g. Soucek and Escoubet (2012)). Having said that, I will continue with more specific comments regarding the content of the current manuscript. There are still concerns regarding the generality of the mapping procedure and the presentation of the method.'*

   **Reply (ref.02.01)**:

   - Thank you very much for the comments. While I understand the wish to include a benchmarking test using the real spacecraft data, I disagree with the suggestion that such a test needs to be included in this manuscript. Of

course, the referee is right in his/her claim, but the goal of the manuscript is to introduce the algorithm for magnetosheath modeling with discussions about merits and demerits from the viewpoint of the algorithm construction. The current manuscript already has a sufficient amount of information and materials. The extension to spacecraft data should be left for a future study, otherwise the manuscript would be significantly long and hard to read. But I happily worked on the further revision based on the specific and technical comments as below.

2. *Specific comments*

*The generalizability of the mapping procedure remains a bit unclear. The phrase "arbitrary shape" has been changed to "non-parabolic shape", but does it also need to be axisymmetric? The discussion reads:*

*line 326: "Our method has the possibility to be extended to three-dimensional, non-axisymmetric modeling by the use of magnetopause normal mapping. It is possible to obtain the steady-state magnetosheath potential in a more general sense without referring to the KF94 solution. [...] Various numerical solvers are known for solving the Laplace equation such as the Jacobi method, the Gauss-Seidel method, and the successive over-relaxation (SOR) method. These Laplace solvers are numerically more expensive than the mapping method, but the computation in 3-D is feasible with the contemporary computational resources."*

*Here, it seems like the Laplace equation needs to be numerically solved for a 3D non-axisymmetric magnetosheath. But I thought your method was to use the analytic expressions from the KF solution and map them onto a magnetosheath with new boundaries. Is this not possible in the non-axisymmetric case? If so, this is quite a crude restriction which should be noted (perhaps in the introduction and/or around line 155). Staying on the topic of the generality of the method, the following sentence is a bit strange:*

*line 170: "We use a specific exponent for the Shue model (with an alpha exponent of 0.5) in an effort to show that the analytic model is 'simple'. The solar wind conditions for which this exponent is applicable is not often encountered"*

*This is a direct response to a previous referee comment. The impression is that you are only showing that the model is simple in a special case which is rarely encountered. With this result, you cannot claim that the general method is 'simple'. Thus, this sentence weakens your argument that the method is simple and/or computationally inexpensive. To improve credibility, would it be possible to give the results with a general alpha exponent?*

**Reply (ref.02.02)**:

- I see that the text in the revision 1 (line 326 and 170) is confusing. I added a section 5 "Discussion" in the revision 2 (page 16, line 311), which shows an application of the method to different exponents of the Shue

model, the approach of generalizing the method to the three-dimensional magnetosheath, and discussion on the different approaches.

3. *The methods section still seems unnecessarily lengthy (compared to the scientific contribution of the study), since the same set of equations are repeated twice with only some changes in the notation. However, if the authors after thorough consideration regard all details as necessary, it can be included as-is.*

**Reply (ref.02.03)**:

- I find the manuscript concise enough, i.e., it is as compact as possible and the readers can still reproduce the results from the information shown in the manuscript.

4. *I have a number of suggestions regarding the figures which might give them a more solid impression:*

*The figure titles are inconsistent – for example, in Fig. 11 the titles describe which functions are plotted and in Fig. 2 the titles refer to the grid and mapping method. The point of this study is that Fig. 11 (left panel) is different from Fig. 2 (right panel), so the 'structure' of the figures should be similar and Fig. 2 should be clearly referred to when discussing Fig. 11.*

*Instead of referring to the figure panels as left/right, why not introduce subfigures (e.g. Fig 2a)? I also suggest to add colorbars so that absolute numbers can be compared between the results of the different methods. In addition, the captions could probably be more informative.*

*It would be nice to have figures that should be compared with eachother side by side (e.g. Fig. 2 (left panel) vs Fig. 2 (right panel) vs Fig. 11 (left panel)), but I understand that this might not be reconcileable with the order in which they are referred to in the text. However, as stated above, there can be more references to the figures (e.g. Fig. 2 vs Fig. 11) when making comparisons.*

*Maybe combine Fig 1 and 3 to facilitate the comparison (keep all plots but make 2x2 subfigures).*

**Reply (ref.02.04)**:

- Figure 2 caption (page 6): "Note that the same function is plotted here for different mapping methods. Contours represent the velocity potential normalized to the solar wind, $\Phi^{(\mathrm{vel})}/U_x$, which is in units of the planetary radii. The color range is chosen for the visual demonstration purpose from $0.2R_{\mathrm{E}}$ to $90R_{\mathrm{E}}$ (left panel) and from $2R_{\mathrm{E}}$ to $200R_{\mathrm{E}}$ (right panel)."
- Figure 11 caption (page 15): "Note that two different functions are plotted here for the same grid and mapping method. The color range is chosen for the visual demonstration from $5.5R_{\mathrm{E}}$ to $314R_{\mathrm{E}}$ (left panel) and from $0R_{\mathrm{E}}$ to $15R_{\mathrm{E}}$ (right panel)."
- Figure 12 caption (page 15): "The color range of the magnetic potential is from $-35R_{\mathrm{E}}$ to $348R_{\mathrm{E}}$."

5. *On line 123, the reader might ask: You say that Soucek et al (2012) were able to avoid the problem, so why is your orthogonality needed?*

**Reply (ref.02.05):**

- We added a paragraph explaining the drawback of the Soucek method (page 5, lines 126–133):

  "Although the method introduced by Genot et al (2011) and later adapted by Soucek et al. (2012) is computational less expensive than global magnetosheath simulations, the density and velocity fields from the bow shock to a given point in the magnetosheath still needs to be computed along the streamline in an incremental way. Moreover, the Rankine-Hugoniot relations need to be solved before calculating iteratively the streamline, the flow velocity vector, to track the plasma density flow velocity along the streamline. Naturally, the uncertainty in this calculations depends on the step size (larger uncertainties for larger step sizes) and the errors accumulate along the streamline. The method introduced in this work overcomes this issue by constructing a magnetopause-normal grid system such that computational efforts are improved (no need to solve the Rankine-Hugoniot relations and the error does not accumulate in the flank region)."

6. *Technical corrections*

   *Text*

   - *u and v are introduced on line 50 but defined/explained on line 72-73. Consider defining them where they are introduced.*
   - *The stream function should be mentioned closer to Eq (9).*
   - *Line 108-110 and line 116-118 are almost the same sentence, a bit repetitive.*
   - *line 323-324: references are in the wrong format (parentheses).*

   *Equations*

   - *Eq (39): Parentheses in the denominator that should not be there.*
   - *Eq (41): Are $e_{mp,x}$ and $e_{mp,y}$ the x and y components of $e_{mp}^{(k)}$? If so, they should have the superscript (k).*

**Reply (ref.02.06):**

- Definitions of the shell and connector variables. Done. Page 2, lines 50–51.

- Stream function. Done. Page 3, lines 69–70.
- Soucek and Escoubet (2012). I leave as is. Both are necessary.
- Parentheses for reference. Done. Page 18, lines 369–370.
- Parenthesis in Equation (39) corrected (page 12).
- Superscript (k) in Eqs. (41) and (42) added (page 12).